# Substance Use and Addiction in Athletes: The Case for Neuromodulation and Beyond

**DOI:** 10.3390/ijerph192316082

**Published:** 2022-12-01

**Authors:** John W. Dougherty, David Baron

**Affiliations:** 1Department of Psychiatry and Behavioral Sciences, Johns Hopkins University School of Medicine, Baltimore, MD 21205, USA; 2Office of the President, Western University of Health Sciences, Pomona, CA 91766, USA

**Keywords:** athletes, substance use, addiction, transcranial magnetic stimulation (TMS), transcranial direct current stimulation(tDCS), ketamine

## Abstract

Substance use, misuse and use disorders continue to be major problems in society as a whole and athletes are certainly not exempt. Substance use has surrounded sports since ancient times and the pressures associated with competition sometimes can increase the likelihood of use and subsequent misuse. The addiction field as a whole has very few answers to how to prevent and secondarily treat substance use disorders and the treatments overall do not necessarily agree with the role of being an athlete. With concerns for side effects that may affect performance coupled with organizational rules and high rates of recidivism in the general population, newer treatments must be investigated. Prevention strategies must continue to be improved and more systems need to be in place to find and treat any underlying causes leading to these behaviors. This review attempts to highlight some of the data regarding the field of substance misuse and addiction in the athletic population as well as explore possible future directions for treatment including Neuromodulation methods and Ketamine. There is a need for more rigorous, high-quality studies to look at addiction as a whole and in particular how to approach this vulnerable subset of the population.

## 1. Introduction

The use and abuse of drugs, primarily alcohol, by athletes is common. Going out for a drink after competition or training is a part of the culture of many sports. Athletes drink and use drugs for several reported reasons; to socialize, self-medicate pain/anxiety, and help falling asleep. When using is intended to improve athletic performance or mask banned substances, it is considered doping. It is thought that athletes’ attitudes towards drugs are heavily influenced by the culture of sports [1].

Alcohol, nicotine, cannabis, stimulants and prescription opioids are the most commonly used substances among elite athletes but overall consumption is lower in professional sports than in the general public [2]. Use of stimulants, prescription opioids and smokeless tobacco products has a higher prevalence in this subset of the population and use of steroids and alcohol along with smokeless tobacco was more commonly used in collegiate athletes compared to their non-athlete counterparts [3,4,5]. Collegiate student-athletes drink more drinks per week [6,7], drink more frequently, consume larger amounts often in correlation with level of athletic involvement [4,8,9,10], and are more likely to drink for social reasons [4,8,11,12]. Involvement in athletics appears to be inversely related to cigarette smoking and illicit drug use [13,14]. In adolescent and young adult athletes, a systematic review found 82% of included studies showing a positive relationship between alcohol use and sports participation and 50% of studies found negative association between participation in sports and marijuana use [14]. Another systematic review indicated higher levels of alcohol use and violence in the athletic population compared to non-athletes [15]. Several studies correlate athletes on teams with higher alcohol use compared to individual sport athletes but this is not consistent among all sports with one study showing male hockey and female soccer athletes more likely to report substance use and male basketball and cross country/track athletes reporting lower levels [16,17,18]. White athletes have a positive correlation with alcohol use whereas black athletes were found to have the inverse relationship [19]. Athletes are less likely to use prescription drugs non-medically with the exception of stimulants but male athletes, athletes with injuries and male athletes with injuries were at greatest risk of non-medical use of prescription opioids [20,21,22]. Males who participated in organized sports were more likely to be prescribed an opioid in past year, had higher odds of misusing and great chance of using to get high compared to non-athletic males but less risk of heroin use [23,24,25]. Opioid use over an NFL career is estimated to be around 52% with 4% using at any given time, whereas one-quarter to one-half of high school athletes have used nonprescription opioids with a lifetime opioid use between 28 and 46% [5,26]. A systematic review found that marijuana use had replaced tobacco use as the second highest used drug among athletes and others suggested one in four athletes have used marijuana recently or within the past year [27,28,29].

## 2. Methods

The major bibliographical research for this paper was performed by using a search strategy exploring PubMed, Google Scholar as well as PsycNet (accessed 19 January 2022 to 9 May 2022). The database searched using multiple keyword combinations including the following words: athletes, addiction, substance use, substance misuse, substance use disorder, alcohol use, alcohol use disorder, drug abuse, alcohol dependence, cannabis, marijuana, cocaine, tobacco use, nicotine, smoker, opioid use, opioid use disorder, transcranial magnetic stimulation (TMS), theta burst, direct transcranial stimulation (tCDS), neuromodulation, ketamine, substance use treatments, substance use disorders, biological passport, review, systematic review and depression. Only studies published in the English language and between 1991 and present were included due to the searches primarily presenting data from that point on in relation to search results requested.

## 3. Where Are We Now?

Currently, good data exist related to substance misuse and addiction in adolescent and college athletes primarily with the predominant substance being alcohol to date. In this review, an attempt will be made to explore the literature related to some of the most common substances of abuse with an emphasis on alcohol, opioids, nicotine, cocaine and marijuana. Steroids along with other stimulants will not be emphasized. There is limited information in relation to treatment of addiction in athletes which will be a primary focus of this paper. Most of the discussion will focus on approaches in the general public with an attempt to project this information onto the athletic population in these cases. One must remember that substance use in athletes may be correlative with traditional uses but may also primarily involve using substances with the intention of improving athletic performance or masking banned substances, known as doping. This behavior has been traced back to ancient Greece. Over the past 45 years, great progress has been made regarding anti-doping policies but yet it continues to play a role at all levels of sporting competition. The recent doping scandal at the 2022 Beijing Games reminded the world of sport, doping has not gone away. What has become evident is that doping practices have evolved from the use of anabolic steroids to improve strength, to masking agents to cover banned substances and strategies to improve endurance and speed-up recovery time from injury and overtraining. Doping has been used to extend athletic careers, as compensation for professional athletes has soared over the past few decades. The core tenets of doping are well articulated by WADA, the World Anti-Doping Agency, the leading global ant-doping agency. These include (1) safety and well-being of the athlete, (2) fair competition and (3) integrity of sport.

Treatment of substance use disorders is broken down into two phases. The first phase consists of prevention which includes psychoeducation in the hopes of providing enough information to sway individuals from using substances initially and secondarily identifying athletes having difficulties with misusing or abusing substances in an attempt to intervene before the behavior progresses to a clearly defined addiction or substance use disorder. It is essential to provide psychoeducation regarding the signs and symptoms of substance misuse along with the dangers associated with their use [30]. To date, the athletic realm has implemented screening tools in an attempt to help prevent progression with preseason physicals, urine drug screens and more recently biological passports. Preseason physicals if done properly can be a first line of defense for picking up cues from athletes regarding their risk for having or developing an addiction. Random drug screens attempt to dissuade athletes from using substances by linking positive test results to negative consequences including disqualification from competition. If found positive, these opportunities may allow the athlete the ability to gain further education on substance use and seek treatment if deemed necessary. It is unclear if random drug testing effectively reduces the use of substances of abuse or if those taking the tests have found ways to circumvent the testing due to its limited frequency [31]. Questions still remain about how to make this process improved but increased frequency has been suggested [31]. Biological passports collect certain biological markers within an athlete’s urine and blood typically effected by doping and compare them in that athlete over the course of time. Passports consist of three parts: endocrinological, steroidal and hematological. These are used primarily when looking at performance enhancing drugs including human growth hormone (HGH), anabolic steroids, Erythropoietin (EPO) among others, as opposed to most of the traditional substances of abuse such as alcohol, marijuana and opiates [32,33]. Passports have intra and inter variability in the realm of hematological markers leading to confounders and demonstrating a need to further improve this testing method in the future [34].

Once an addiction is identified, screening for a co-occurring mental health disorder should be performed due to the increased frequency in those using substances and the possibility that they may be attempting to self-medicate. In 2017, there were 20.3 million individuals with substance use disorders in the U.S., 37.9% or 7.7 million had co-occurring mental health disorders [35]. These numbers may be underreported especially in the athletic population where there is an increased stigma related to mental health compared to the general population. Elite athletes experience a fairly comparable risk to mental health disorders in relation to non-athletes except in cases that may involve situations such as retirement and injury [36,37,38]. Understanding an athletes’ reason(s) for use may help further elucidate what possible modalities of treatment may be the best fit for the circumstance. More mental health resources need to be allocated to teams and sports including access to a sports psychiatrist when cases present themselves so there is no gap in treatment. 

Within the context of substance use treatment, there are several evidence-based medications and therapy methods that have been found to be effective for these disorders. Out of the present studies, very few have explored therapeutic techniques in athletes. Motivational interviewing (MI), Cognitive behavioral therapy (CBT) and Contingency Management (CM) are implemented to increase motivation to decrease use and ultimately change their behaviors. There is no reason to believe these techniques and variations of such would not be successful in athletes but more studies are needed. In athletes, a few studies looked at spit tobacco use by implementing dental exams with subsequent counseling by the dental technician if screening for nicotine use was positive. In one study, tobacco use decreased by about 15.5% over a 10-year period incorporating these methods [31,39]. As little as one brief intervention session with athletes demonstrated decreased alcohol use, frequency and alcohol related consequences three months later [40]. Another study demonstrated that online personalized drinking feedback (PDF) coupled with athlete specific information led to a lower peak blood alcohol concentration (BAC) at 6-month follow-up along with in-season athletes reporting less drinks per week than those in the control group [41]. Other online programs have found that providing web-based feedback or online modules in college athletes could lead to significant reductions in drinking, as well as improvement in assessment of social norms related to drinking [42,43]. Alcoholics anonymous (AA) and Narcotics anonymous (NA) meetings along with finding a sponsor are effective methods as well with no data in the athletic world at this time. For cases where substance use leads to significant impairment in daily functioning and they cannot come off substances safely without risks of medical complications, those individuals are referred to either intensive outpatient programs (IOP) or more likely inpatient rehabilitation centers or detoxification centers offering anywhere from 7 to 30 days of treatment. Most athletes would struggle with the above options more than the average patient due to fears of confidentiality, stigma within their particular sport as well as repercussions related to their struggles becoming public ultimately leading to fear of punishment, feelings of letting teammates down as well as being perceived as weak [44].

Although these therapies can be effective, often times they are not successful alone to maintain abstinence from substances. In these cases, medication interventions may need to be implemented as part of the treatment plan. One must keep in mind that no studies have looked at the role of these medications in athletes or the impact they may have on performance. Most medication options for substance use disorders function to target cravings or help to prevent withdrawal. Currently, the main disorders treated by medications primarily involve nicotine, alcohol and opiates. In regard to nicotine, the primary treatments involve nicotine replacement options in the form of gums, patches, lozenges or sprays to help treat cravings, bupropion and varenicline. Bupropion is an antidepressant that works on norepinephrine and dopamine with some activity at nicotinic receptors. Varenicline is a partial nicotine receptor agonist. All of these medications promote smoking cessation. For alcohol, the four FDA approved mainstays of treatment are naltrexone/vivitrol, acamprosate to help with reducing cravings and disulfiram which impairs the body’s ability to break down acetaldehyde alcohol leading to physical illness as a deterrent from drinking which is not used often in current medicine practice. Disulfiram can be an effective tool when used properly but has been somewhat phased out as the drug of choice. Naltrexone is a newer medicine, an opioid antagonist which has been found to be effective in reducing cravings for alcohol and opiates and prevents the euphoric effects associated with drinking. An injectable form, Vivitrol, is a once-a-month shot given to help prevent relapse and removes the user error from the equation to increase chances of success in a population with high recidivism. It is unclear how exactly acamprosate functions to decrease cravings but does have glutamatergic excitatory activity. Two other medications used in the opioid population include methadone (a mu-opioid receptor agonist) and suboxone (buprenorphine/naloxone) is a partial mu opioid receptor agonist combined with an opioid antagonist. Subclade is a long-acting injectable form of buprenorphine (without naloxone). These treatment methods were designed for prolonged use to help prevent withdrawal and cravings regarding opiates. Methadone requires going to specific treatment centers to get dosing and can be associated with sedation and cardiac complications which would not be compatible with athletes’ lifestyles. Suboxone, which is buprenorphine and naloxone (a rapid acting opioid antagonist to prevent abuse amongst users), is thought be better tolerated overall. Both methadone and buprenorphine are prohibited by WADA (World Anti-Doping Agency) in competition so is likely not an option for active athletes. 

Despite the treatments above, they are limited and flawed. In a multisite, randomized controlled study, the rate of successful outcomes after 12 months with suboxone was under 50% and had a relapse rate of 57% whereas those treated with vivitrol had a relapse rate of 65% [45]. All these treatments still have plenty of room for improvement, only offer limited disorders assistance and do not even begin to explore the athletic population and their specific needs. This is coupled by the fact that athletes do not typically like to take medications as they tend to be young and healthy and are quite fearful of side effects. Other substances outside of nicotine have even less data supporting any type of medication treatment including stimulants, cannabis and cocaine amongst others. 

## 4. Where Do We Go from Here?

To date, the literature is filled with a limited number of high-quality systematic reviews in the field of addiction related to athletes and these studies mainly examine a small number of studies [14,15,26,27,29]. Several studies have also indirectly compared athletes and non-athletes looking primarily at collegiate athletes in the U.S with the most recent reporting lower annual use of substances such as alcohol, marijuana, amphetamines, cigarettes, cocaine, ecstasy and anabolic steroids compared to non-athletes [31,46]. There appears a need for more rigorous high-quality studies looking at direct head-to-head comparisons between athletes and non-athletes in the field of addiction with an emphasis on treatments.

Athletes as a whole are more likely to accept treatment that does not require a daily medication and maintain a preference to avoid any treatment that may contain side effects that can affect performance. These preferences may align more with newer alternative treatments that include various forms of neuromodulation. Two types of non-invasive neuromodulation to investigate in athletes with addiction include transcranial magnetic stimulation (TMS) and transcranial direct current stimulation (tDCS) [45,47]. TMS is a brain stimulation technique targeting the dorsolateral prefrontal cortex (DLPFC) where a metal coil is positioned against the scalp to generate rapidly alternating magnetic fields that then pass through the skull and depolarize neurons in the particular area. The exact mechanism of action is still unknown but it is believed to contribute to long term excitation and inhibition of neurons in certain portions of the brain. TMS has been effectively used since 2008 for major depressive disorder in those individuals who have failed one antidepressant and 2018 for obsessive compulsive disorder. Side effects are self-limiting to headache and scalp pain with a slight increased risk of seizure of up to 0.5% [48]. Transcranial direct current stimulation applies a low-intensity direct current through two electrodes over the scalp producing an electrical field in the brain leading to neuronal changes powered by a battery-operated machine. Low-intensity current is given for 30 min per session and the number of sessions can vary. Common side effects are limited and can involve nausea, dizziness, headache and skin irritation. One potential benefit of this type of treatment is that it can allow the possibility of home treatment providing flexibility and protecting confidentiality. The downside of this treatment is that it is not currently FDA approved and its use is considered investigational or experimental at this time in the United States. More randomized controlled studies are needed to be performed to demonstrate the potential true benefit of this treatment especially in sports population. Many athletes may be hesitant to using any treatment that was not FDA approved due to fears of violating the rules by which their sports are governed. 

When discussing TMS, one must consider the current treatment protocol and whether that might be ideal for an athlete’s availability and timeline. TMS treatment as typically constituted for the depression protocol can last 30–40 min, five days a week for about 6–7 weeks. This immense time commitment might be another roadblock for those seeking rapid recovery in the context of the sporting world pressures. To combat this issue, theta burst TMS may be another alternative that uses a higher frequency to produce shorter treatment sessions of 3 to 10 min. This allows for more succinct treatments and therefore more treatments within the same day therefore decreasing the overall timeline for those athletes who have limited time outside their responsibilities to their sport especially during the season. In the Stanford Accelerated Intelligent Neuromodulation Treatment study (SAINT), 19/22 depressed patients (90.5%) met remission criteria after 10 treatments over the course of five days providing significant hope for those athletes at least who have a mental health disorder and potentially a greater increase in success in those with co-occurring substance use [49]. While WADA does not currently prohibit the use of TMS, there is some question about whether there should be more oversight in relation to any possible performance enhancing effects as classifying new treatments as banned becomes more challenging and has sparked debate within the world of sport [50,51].

In this section, we will attempt to present some of the current data looking at TMS, tCDS in addiction treatment in an effort to project the positive prospects onto athletes due to a lack of data presently available related to athletes specifically. It is important to relay that none of these studies presently look at athletes but this review attempts to lay out some data to support further evaluating such treatments in this distinct population. To date, TMS consists mostly of studies involving alcohol, cocaine and nicotine with very few looking at methamphetamine use which will not be discussed at this time. There are only a handful of reported studies looking at opioid use disorder treatment. For alcohol specifically, several studies demonstrated in the general population a decrease in craving but not cue-induced alcohol craving [52,53,54,55], whereas others did not affect craving at all [56,57]. Data for reduced alcohol intake have been mixed with one study demonstrating reduced alcohol intake without craving change and another failing to reduce craving or alcohol intake [58,59]. All of these studies consisted of a small number of patients from 8 to 30 in total so larger more robust studies are still needed to be performed to explore this potential benefit. Transcranial direct current stimulation consists of two studies in relation to alcohol with one showing a 27.3% relapse rate compared to 72.7% in control group at 3 months and 50% compared to 80% in sham group at 6 months [60,61]. With theta burst, there is some data to support impaired alcohol intake as well as the potential modulation of signals induced by drugs in cortex areas involved in dependence [62,63]. In the transcranial direct current studies, the results have been tepid at best with the most recent metanalysis revealed small positive effects on alcohol craving and consumption which contradicted a previous metanalysis [64,65]. One RCT performed recently demonstrated higher rates of abstinence in those treated with tDCS compared to other conditions but only for two weeks post rehabilitation [66].

For nicotine use, several TMS studies demonstrate reduced craving and consumption of cigarettes one study in schizophrenic patients, showed cessation of craving but did not affect abstinence rates, potentially confounded by the co-morbid mental health disorder and another found reduced consumption of cigarettes but no effect on craving [45,52,67,68,69,70]. Short term TMS found no effect on craving [71]. Several others looked at abstinent smokers and found either reduced craving, or improved success rate of abstinence by reducing risk of relapse by 3-fold [45,72,73]. The most recent study, a multicenter RCT demonstrated reduced consumption and cravings [74]. In the only study to look at theta burst in nicotine users, abstinence rates were increased three months post treatment but cravings were unchanged [45,75]. A metanalysis involving twelve studies looking at tDCS on symptoms of nicotine dependence demonstrated significant positive changes in smoking intake and craving related to cues [76]. The most recent study looking at smokers not looking to quit were treated with tDCS which cut their cravings by 50% but intake remained the same [77].

TMS studies looking at cocaine primarily all demonstrated decreased craving compared to the control group [52,78,79,80,81,82,83]. Several demonstrated reduced intake and craving and a single study looked at treatment of 11 weeks leading to an elongated latency to the first relapse [52,81,82,84]. Finally, one single theta burst study performed three sessions a day for 10 days and demonstrated a reduction in overall days cocaine was used by 70% and a 78% reduction in weekly cocaine consumption spending based in dollars [85]. TMS looking at opioid use disorder treatment remains quite limited. One of the only studies that was randomized to explore this possibility looked at 20 heroin using males and found a decrease in overall cravings, another demonstrated reduced cue induced cravings but was not statistically significant and one study related to tDCS found similar results [86,87,88,89]. Two studies utilizing tDCS looked at opioid use and pain in those who underwent a total knee arthroplasty with both suggesting decreased pain medication use but areas of treatment conflicted [90,91]. This may be an important area to focus on for the cases of injured athletes with their injury playing a role in the development of their misuse. Cannabis has very little data related to TMS with only one study demonstrating decreased cravings but the study was open label and a very small sample size, whereas another did not see a decrease with one treatment [92,93]. Finally, one tDCS study demonstrated decreased cravings as well in the right/left anodal DLPFC group [94].

One other promising avenue for potential future treatment may involve the use of ketamine for substance use issues in athletes. Ketamine is an anesthetic drug that has been demonstrated to be effective in the treatment of pain. It was discovered to have antidepressant properties at low doses since an initial study in 2000 demonstrating positive results with a single infusion that has been expounded upon by several RCTs since that time showing sustained antidepressant effects with repeat dosing [95,96,97]. Ketamine is an NMDA receptor antagonist but its true mechanism of action has still not been completely elucidated. Recent studies suggest it amplifies targeting of rapamycin (mTOR) signaling as well as increasing brain derived neutrophic factor (BDNF) which promotes neuronal survival among other possible functions [98]. Ketamine is FDA approved for the purposes of anesthesia and an S enantiomer version of the drug known as Spravato (esketamine) has been approved for depression. Intravenous racemic ketamine (mixture of R and S enantiomers), the most commonly used form for treatment, has not to date been approved for depression and neither version is approved for substance use disorders. Typically, this treatment involves six 40 min infusions over the course of 2–3 weeks. Currently, there are limited data to support the potential benefits in alcohol, cocaine and opioid use disorders. Out of four main studies involving alcohol, ketamine patients were found to have increased abstinence, delayed time to relapse, decreased heavy drinking days, improvement in cravings and when used after brief retrieval of maladaptive cue-alcohol memories demonstrated a reduction in the reinforcing effects of alcohol [99,100,101,102,103]. Cocaine studies demonstrated a 60% increase in motivation to quit, reduction in cue induced craving, a decrease in self-administration as well as upwards of almost 50% of ketamine patients remained abstinent the last two weeks of one study and were over 50% less likely to relapse compared to the control group [104,105,106]. One study related to marijuana found a decrease in cannabis use frequency [103,107]. Despite the limited nature of this literature, the data presented so far may be a promising avenue to explore in a population in need of better treatment options. 

Athletes may be apprehensive to engage in ketamine treatment due to the fact that it is not FDA approved for substance use disorders as well as the stigma associated with its use. The potential pros of treatment if efficacious would be the rapid nature of treatment and the limited side effects that traditionally are self-limited to brief fatigue, dizziness, nausea, headache and dissociation. Athletes who are triggered to use substances due to underlying mental health disorders may benefit even more from this treatment since in some cases positive antidepressant effects can occur even with one treatment or within a few weeks. Athletes may be more prone to look towards Spravato since it is FDA approved but the requirements involving monitoring and a rigorous treatment schedule that may not fit into an active athlete’s routine and desire for confidentiality by having to sit in a waiting room with other patients. Intravenous ketamine infusions have a lot of data to support them and would allow more flexibility in regards concerns above. As of now, Ketamine is not on WADAs banned substance list and is not something that is screened for on a urine drug screen allowing the possibility of its use. Ketamine has at times been a substance of abuse but small subanesthetic doses used in medically monitored settings have demonstrated a low risk for this outcome. Athletes could feel comforted in the fact that there is no daily oral pill and any side effects including the active drug in their system is transitory around the day of treatment predominately which makes this a very appealing potential option. If a series of treatment was successful, they may only have to ultimately received an infusion treatment a month on average. Just with TMS, there still needs to be more high-quality well controlled studies involving substance use and more specifically the unique population of athletes. There is a need to control as many variables as possible and the athletic community must continue to find ways to encourage their own to speak up in their struggles without fears of retaliation or punishment. Prospective randomized controlled studies will allow providers to have more confidence in these treatments.

## 5. Conclusions

Overall, it is a common belief that substance abuse and addiction likely occur at a lower rate in athletes compared to the general population [2]. Competing at a high level appears to be somewhat protective in some areas of sport outside of the concept of doping. It can be surmised that this paradigm may change after certain events such as injury and retirement which may lead to more vulnerability. Some anecdotal reports have demonstrated this point but still more work needs to be done in the area. Prevention is crucial in the process of reducing the risk of addiction with education, identification and implementing testing to trigger negative consequences for those who are caught using. Further examination of these policies may be warranted to balance the deterrent aspect with the idea of incorporating a welcoming environment where athletes feel comfortable seeking help. Most of the available literature primarily looks at substance use in adolescent and college athletes with more emphasis on alcohol predominately and is limited in relation to treatment modalities. Current existing medications have variable success at preventing relapse providing rationale to investigate alternative methods. A goal in the treatment of athletes would be to find either new medications without side effects such as sedation, weight gain or cardiac effects or non-medication options including neuromodulation techniques discussed above that can provide benefit without any daily side effects. Athletes need to feel confident that a treatment will not cause impairment or violate any anti-doping policies. TMS, tDCS and ketamine provide promising results for the future of addiction treatment as a whole. There is a still a need for more randomized controlled trials looking at these newer approaches to the field of addiction with larger sample sizes and studies must find ways to limit variables more specifically in the realms of cravings, remission length, polysubstance abuse, area of treatment in cases of TMS and tDCS and co-occurring mental health disorders as well as in athletics looking at specific sports and gender while at the same time expounding on each of these over time to provide more specific guidance.

## Data Availability

Not applicable.

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
