# Peer review of "Substance Use and Addiction in Athletes: The Case for Neuromodulation and Beyond"

_ijerph, 2022, doi:10.3390/ijerph192316082_

Round 1
Reviewer 1 Report
The paper is very interesting and touches one of the main problems in toxicology: substance misuse in athletes. In order to improve the quality of the paper I have the following suggestions to authors:
line 16: "This review attempts to highlight the limited data " - there are a relatively large number of data and reports about substance misuse in the field of sport, therefore, the term "limited data" should be replaced with "insufficient" (or other term) if authors consider that there should be more, but "limited" might induce the feeling that there was not made a proper literature research.
line 19: is really ketamine a good substance to be recommended to athletes? If the answer is yes, a very good literature research should be done and a very good explanation should be provided.
line 30, 34: not every reader is familiar to CAGE and TMS - remember that this paper is for those that try to learn something - therefore, every such term should be defined at the first appearance in the text or at the end in an abbreviation list (or even both).
line 49: "Currently, there is very little information available regarding substance misuse and addiction in athletes" - there are a huge amount of data or reports available. "Insufficient" may be used instead of very little. Maybe the authors should search for supplementary information not published on Pubmed.
line 63: WADA means? for those that are not familiar with this terms? Through all the article every term should be defined as I mentioned earlier.
line 139: it is worth to mention that disulfiram is an obsolete medication, and naltrexone is the new trend in reducing alcohol use relapse.
line 226: 21]. should be replaced with 21],
Unfortunately, the paper discusses mainly nicotine and alcohol addiction, with some mentions about cocaine and opioids. These are not the only/main substances used for performance increase in athletes, therefore a change in the title could be useful.
Otherwise, the paper is well written and well documented.
Author Response
Please see uploaded revision of paper with major changes related to several points made

Reviewer 2 Report
This study reviewed literature of different treatments for substance abuse on athletes, the authors searched literatures in MEDLINE and PubMed by these key words: athletes, addiction, substance use, substance misuse, alcohol use, drug abuse, alcohol dependence, cannabis, marijuana, cocaine, tobacco use, nicotine, smoker, opioid use, transmagnetic stimulation, theta burst, direct tanscranial stimulation, neuromodulation, ketamine, substance use treatments, substance use disorders, biological passport and depression. After reviewing, the authors argued that the effectiveness of neuromodulation methods and ketamine should be widely examined in future.
Indeed, substance use and abuse in athletes is vey important issue, so that we need to pay more attentions to examine different form treatments about substance use disorders on athletes. However, some limitations and rationales should be considered in further:
The first, the introduction is too brief to understand the focuses in this review. Regarding to research goals of the present study, the evidences of treatments for substance use disorder or substance use or abuse in athletes which one was main goal in this study? I have confused that. Because of your research goals will match the key words for searching literature. Please to have more illustrations on the research goals, for example, the authors shifted from assessment of drug use to neuromodulation in athletes in section of Introduction.
The second, in method, the authors did not provide the intervals of literatures (from which year to which year?). Then, please to show the criteria of selecting literature, for example, recruit and excluded criteria. The studies of ketamine treatments were reviewed did not recruit athletes in above studies, therefore, it could not be appropriate to support the effectiveness of ketamine use.
The third, the authors argued several medications could not be available for athletes despite these drugs could reasonably reduce the extents of craving. They began to introduce the potential effectiveness of TMS and tDCS. Then, use of ketamine was introduced in last section. I have confused why the authors mentioned about ketamine on treatment of SUD for athletes without any reference or citations.
The forth, some studies supported the effectiveness of TMS or tDCS, however, the participants in above studies were not pure athletes without substance use disorders.
Thanks for your patience, hope these comments are helpful to the authors.
Author Response
made major changes related to the comments below, please see attached revised copy

Reviewer 3 Report
Substance Use and Addiction in Athletes: The Case for Neuro-1 modulation and Beyond (IJERPH 1687033)
The manuscript reviews some of the literature on substance use and addiction in athletes and then pivots to a review of newer treatments for substance use problems and conditions that can contribute to self-medication (e.g., depression) such as transmagnetic stimulation, theta burst, direct transcranial stimulation, and therapeutic use of ketamine.
The main topic of the article is interesting and I am not aware of a prior review of these newer therapeutic methods as they could be applied to athletes. There are also some major gaps in the literature review and some statements in the manuscript appear to be inaccurate.
Comments below may serve to strengthen the manuscript:
- A primary concern is that the manuscript in some places appears to be unaware of the existing literature. The manuscript indicates that “Currently, there is very little information available regarding substance misuse and addiction in athletes.”
I think there are actually quite a few articles on this topic, some of which I list at the end of the review (note that this is not intended to be a comprehensive list). There are even some prior reviews of this literature (Kwan et al., 2014; Sønderlund et al., 2014) . I was familiar with two of these articles and found them on Google Scholar and then quickly identified several more by looking at articles that cited those.
The manuscript notes that a search was conducted on Medline. It may be worth conducting a more thorough search using multiple search engines (e.g., PubMed, Web of Science, PsychInfo). As it stands, the current manuscript does not review the existing literature in a substantive way.
If the intent is simply to review newer treatments for athletes, that should be explicitly stated and at least a brief (and accurate) summary of the existing literature on substance use in athletes is needed.
- Related to this, some speculative statements in the manuscript could be bolstered by citations of the exiting literature (e.g., “Overall, it is a common belief that substance abuse and addiction likely occur at a lower rate in athletes compared to the general population”).
- It is helpful to spell out abbreviations such as TMS the first time they are used.
- Although generally well-written, the manuscript needs a careful editing (e.g., “Athletes must also made aware”, “merely covers up acutely problems”, starting sentences with capital letters [e.g., “whereas others did not affect craving…”]).
Articles on substance use in athletes (apologies for the formatting):
Substance use among college athletes: A comparison based on sport/team affiliation
Ford
2007/1/1
Journal of American College Health, 55: 367-373
Alcohol use among college students: A comparison of athletes and nonathletes
Ford
2007/1/1
Substance use & misuse, 42: 1367-1377
Relationship of high school and college sports participation with alcohol, tobacco, and illicit drug use: A review
NE Lisha, S Sussman - Addictive behaviors, 2010 - Elsevier
Sport participation and alcohol and illicit drug use in adolescents and young adults: A systematic review of longitudinal studies
M Kwan, S Bobko, G Faulkner, P Donnelly, J Cairney - Addictive behaviors, 2014 – Elsevier
The association between sports participation, alcohol use and aggression and violence: A systematic review
AL Sønderlund, K O'Brien, P Kremer, B Rowland… - Journal of science and medicine in sport 2014
Contextual influences and athlete attitudes to drugs in sport
ACT Smith, B Stewart, S Oliver-Bennetts… - Sport management review, 2010 – Elsevier
Nonmedical prescription drug use among college students: A comparison between athletes and nonathletes
JA Ford - Journal of American College Health, 2008
Author Response
made major changes to this paper. Please see attached for all the changes related to the comments below

Round 2
Reviewer 3 Report
Substance Use and Addiction in Athletes: The Case for Neuro-1 modulation and Beyond (IJERPH 1687033.R1)
The manuscript reviews some of the literature on substance use and addiction in athletes and then pivots to a review of newer treatments for substance use problems and conditions that can contribute to self-medication (e.g., depression) such as transmagnetic stimulation, theta burst, direct transcranial stimulation, and therapeutic use of ketamine.
The main topic of the article is interesting and I am not aware of a prior review of these newer therapeutic methods as they could be applied to athletes.
The authors have generally done a fine job revising the manuscript and it has been strengthened as a result.
Two additional minor edits are recommended:
“Athletes are less likely to use non-prescription drugs non-medically with the exception of stimulants…”
Should read “less likely to use prescription drugs non-medically”
Incomplete edit: “Ketamine is an anesthetic drug has been demonstrated…”
Author Response
changed sentence as requested, also added that to the second comment to make it a sensible sentence.
